# Light Intensity and Photoperiod Affect Growth and Nutritional Quality of Brassica Microgreens

**DOI:** 10.3390/molecules27030883

**Published:** 2022-01-28

**Authors:** Kaizhe Liu, Meifang Gao, Haozhao Jiang, Shuying Ou, Xiaopeng Li, Rui He, Yamin Li, Houcheng Liu

**Affiliations:** College of Horticulture, South China Agricultural University, Guangzhou 510642, China; 1836945107@stu.scau.edu.cn (K.L.); gmf@stu.scau.edu.cn (M.G.); jhzh111@stu.scau.edu.cn (H.J.); 20192128001@stu.scau.edu.cn (S.O.); lxpzs@stu.scau.edu.cn (X.L.); ruihe@stu.scau.edu.cn (R.H.); yaminli@stu.scau.edu.cn (Y.L.)

**Keywords:** light intensity, photoperiod, cabbage, Chinese kale, nutrition, antioxidants

## Abstract

We explored the effects of different light intensities and photoperiods on the growth, nutritional quality and antioxidant properties of two Brassicaceae microgreens (cabbage *Brassica oleracea* L. and Chinese kale *Brassica alboglabra* Bailey). There were two experiments: (1) four photosynthetic photon flux densities (PPFD) of 30, 50, 70 or 90 μmoL·m^−2^·s^−1^ with red:blue:green = 1:1:1 light-emitting diodes (LEDs); (2) five photoperiods of 12, 14, 16, 18 or 20 h·d^−1^. With the increase of light intensity, the hypocotyl length of cabbage and Chinese kale microgreens shortened. PPFD of 90 μmol·m^−2^·s^−1^ was beneficial to improve the nutritional quality of cabbage microgreens, which had higher contents of chlorophyll, carotenoids, soluble sugar, soluble protein and vitamin C, as well as increased antioxidant capacity. The optimal PPFD for Chinese kale microgreens was 70 μmol·m^−2^·s^−1^. Increasing light intensity could increase the antioxidant capacity of cabbage and Chinese kale microgreens, while not significantly affecting glucosinolate (GS) content. The dry and fresh weight of cabbage and Chinese kale microgreens were maximized with a 14-h·d^−1^ photoperiod. The chlorophyll, carotenoid and soluble protein content in cabbage and Chinese kale microgreens were highest for a 16-h·d^−1^ photoperiod. The lowest total GS content was found in cabbage microgreens under a 12-h·d^−1^ photoperiod and in Chinese kale microgreens under 16-h·d^−1^ photoperiod. In conclusion, the photoperiod of 14~16 h·d^−1^, and 90 μmol·m^−2^·s^−1^ and 70 μmol·m^−2^·s^−1^ PPFD for cabbage and Chinese kale microgreens, respectively, were optimal for cultivation.

## 1. Introduction

Microgreens have gained considerable attention due to their being excellent sources of various phytochemicals that are important for the health and nutrition of the human body. Brassica microgreens have received much interest for their having higher nutrient content, specifically of glucosinolates and vitamin C, than mature vegetables [1]. The production and accumulation of phytochemicals from microgreens could be significantly affected by light conditions [2].

Light conditions (light quality, light intensity and photoperiod) are one of the most essential environmental factors in modifying plant morphology and physiology through a complete lifecycle [3,4]. Light-emitting diodes (LEDs) have been used to optimize crop production and quality in controlled environments with several advantages over traditional lighting sources.

Some studies have been performed to dissect the role of light quality on nutritional quality and bioactive compound levels of microgreens [5,6], however, there is little information about the effects of different LED light intensities or photoperiods on the growth and nutrition of Brassicaceae microgreens.

Broccoli microgreens cultured under different light intensities (30, 50, 70 and 90 µmol·m^−2^·s^−1^ with red:green:blue = 1:1:1) displayed alterations in growth and phytochemical contents, which indicated that 50 µmol·m^−2^·s^−1^ was the optimal light intensity for the production of broccoli microgreens, while 70 µmol·m^−2^·s^−1^ was more beneficial for phytochemical accumulation. Glucosinolates (GSs), as important, health-promoting compounds in Brassica microgreens, are also affected by light intensity [7]. The content of some individual GSs (e.g., PRO, SIN, GBN, GER and NGBS) decreased significantly as light intensity increased from 30 to 50 µmol·m^−2^·s^−1^, but increased as light intensity increased from 50 to 90 µmol·m^−2^·s^−1^ [7]. In *Brassica oleracea*, total GS content decreased as light intensity increased [8]. Moreover, different intensities (0, 12.5, 25, 50 and 100 µmol·m^−2^·s^−1^) notably affected the morphological characteristic of the sprouts of six types of vegetables [9].

Photoperiod had significant impacts on plant morphology, which is directly related to plant biomass. The biomass of kale increased linearly with photoperiod [10]; higher fresh and dry mass were observed with longer photoperiods, reaching the maximum in a 24-h photoperiod [10]. Different photoperiods (0, 8, 12, and 16 h of light) significantly influenced the appearance of Chinese kale sprouts, but exerted a negligible effect on GS accumulation [11].

This study is to explore the effects of LED light intensities and photoperiods on the growth and phytochemical content of cabbage (*Brassica oleracea*) and Chinese kale (*Brassica alboglabra*) microgreens in order to provide information for high-quality microgreen production in artificial lighting plant factories.

## 2. Results

### 2.1. Effects of Light Intensity on Growth and Quality of Cabbage and Chinese Kale Microgreens

#### 2.1.1. Growth and Biomass

With increasing light intensity, the hypocotyl length of cabbage and Chinese kale microgreens shortened (Table 1). Dry and fresh weights varied significantly among treatments. The fresh weight of cabbage microgreens was the highest under 50 μmol·m^−2^·s^−1^, while the dry weight was the highest under 70 μmol·m^−2^·s^−1^. The fresh weight of Chinese kale microgreens under 50 and 70 μmol·m^−2^·s^−1^ was significantly higher than other treatments. The dry weight of Chinese kale microgreens under 30 μmol·m^−2^·s^−1^ was significantly lower than other treatments (Table 1).

#### 2.1.2. Pigment Content

Below 70 μmol·m^−2^·s^−1^, with increasing light intensity the contents of chlorophyll a, chlorophyll b, total chlorophyll and carotenoid in cabbage and Chinese kale microgreens significantly increased (Table 2). However, at 90 μmol·m^−2^·s^−1^ the pigment content of Chinese kale microgreens was significantly lower than for 70 μmol·m^−2^·s^−1^ (while for both types of microgreens it remained significantly higher than for 30 μmol·m^−2^·s^−1^).

#### 2.1.3. Content of Soluble Protein, Soluble Sugar, Vitamin C and Nitrate

The soluble sugar content of cabbage microgreens increased with increasing light intensity (Figure 1A), and the highest content was at 90 μmol·m^−2^·s^−1^. The soluble sugar content in Chinese kale microgreens increased with the increase of light intensity until peaking at 70 μmol·m^−2^·s^−1^, then decreased at 90 μmol·m^−2^·s^−1^. The soluble sugar content of Chinese kale microgreens was higher than that of cabbage.

The soluble protein content of cabbage microgreens increased with the increasing light intensity (Figure 1B), and the highest content was at 90 μmol·m^−2^·s^−1^. However, the soluble protein content of Chinese kale microgreens was the highest when the light intensity was 70 μmol·m^−2^·s^−1^. The soluble protein content in Chinese kale microgreens was higher than that of cabbage from 30 to 70 μmol·m^−2^·s^−1^, but lower at 90 μmol·m^−2^·s^−1^.

The vitamin C content of cabbage microgreens increased with the increasing light intensity (Figure 1C), and the highest content was at 90 μmol·m^−2^·s^−1^. However, there was no significant difference in vitamin C content of Chinese kale microgreens among treatments.

The nitrate content in cabbage and Chinese kale microgreens was the lowest under 90 μmol·m^−2^·s^−1^ (Figure 1D). In different treatments, the nitrate content in Chinese kale microgreens was higher than that of cabbage. From 30 to 90 μmol·m^−2^·s^−1^, the nitrate content of Chinese kale was 82.1%, 96. 4%, 12. 7% and 8.1% higher, respectively, than that of cabbage.

#### 2.1.4. Antioxidant Content and Capacity

There was no significant difference in DPPH radical-scavenging rates of cabbage microgreens among treatments (Figure 2A), while that of Chinese kale microgreens was significantly lower under 30 μmol·m^−2^·s^−1^ than other treatments.

The FRAP value (content of polyphenol and flavonoid) in cabbage and Chinese kale microgreens under 30 μmol·m^−2^·s^−1^ was significantly lower than other treatments. Flavonoid contents in Chinese kale microgreens were significantly higher than those in cabbage microgreens (Figure 2D).

#### 2.1.5. Heatmap Analysis

The heatmap demonstrates the effects of different light intensities on the growth and quality of cabbage and Chinese kale microgreens. Under 90 μmol·m^−2^·s^−1^, the dry weight, soluble protein, soluble sugar, vitamin C, pigment, polyphenol, flavonoid contents and FRAP value of cabbage microgreens were the highest (Figure 3A), whereas the fresh weight, hypocotyl length and nitrate contents were the lowest.

At 30 μmol·m^−2^·s^−1^ light intensity, the hypocotyl length of Chinese kale microgreens was highest (Figure 3B). At 70 μmol·m^−2^·s^−1^, the hypocotyl length, nitrate and soluble protein content were lower, while the fresh weight, dry weight, soluble sugar, vitamin C, pigment, polyphenol, flavonoid contents, FRAP value and DPPH radical-scavenging rate were higher.

#### 2.1.6. Multivariate Principal Component Analysis

The global comparison of the correlations among all tested phytochemicals in cabbage and Chinese kale microgreens under different light intensities was generated through principal component analysis (PCA) (Figure 4).

For cabbage microgreens, 30, 50, 70 and 90 μmol·m^−2^·s^−1^ light intensities were divided into different quadrants in the PCA biplot, which indicated significantly different reaction patterns (Figure 4A). The first four principal components (PCs) were associated with eigenvalues higher than 1 and explained 88.4% of the cumulative variance, with PC1 accounting for 49.1%, PC2 for 20.6%, PC3 for 11.7%, PC4 for 7.1% and PC5 for 3.8% (Table 3). PC1 was positively correlated to dry weight (DW), chlorophylls (Chl a, Chl b, and Total chl), FRAP, carotenoids (Caro), phenolic (Phe), soluble protein (Pro), flavonoids (Fla), vitamin C (VC) and soluble sugar (Sugar). PC1 was also negatively related to DPPH, hypocotyl length (HL), nitrate (Nit) and fresh weight (FW). Additionally, hypocotyl length (HL), nitrate (Nit), fresh weight (FW), FRAP, dry weight (DW), chlorophylls (Chl a, Chl b, and Total chl), carotenoids (Caro) and phenolic (Phe) positively contributed to PC2, while DPPH, flavonoids (Fla), soluble protein (Pro), vitamin C (VC) and soluble sugar (Sugar) negatively correlated to PC2. Furthermore, the angle (α) between any two vectors described positive (α < 90°), negative (α > 90°) or no significant correlation (α = 90°) between the corresponding parameters. Strong positive correlations were found between dry weight (DW), soluble protein (Pro), soluble sugar (Sugar) and vitamin C (VC), as their angles were less than 90°.

For Chinese kale microgreens, different light intensities were divided into different quadrants in the PCA bi-plot, which indicated significantly different reaction patterns (Figure 4B). The first three principal components (PCs) were associated with eigenvalues higher than 1 and explained 88.7% of the cumulative variance, with PC1 accounting for 49.6%, PC2 for 23.1%, PC3 for 16.0%, PC4 for 5.0% and PC5 for 2.3% (Table 3). PC1 was positively correlated to fresh weight (FW), vitamin C (VC), dry weight (DW), chlorophylls (Chl a, Chl b, and Total chl), FRAP, soluble sugar (Sugar), flavonoids (Fla), DPPH, phenolic (Phe) and carotenoids (Caro). PC1 was also negatively related to hypocotyl length (HL), soluble protein (Pro) and nitrate (Nit). Additionally, hypocotyl length (HL), soluble protein (Pro), nitrate (Nit), fresh weight (FW), vitamin C (VC), dry weight (DW), chlorophylls (Chl a, Chl b, and Total chl), FRAP, soluble sugar (Sugar) and flavonoids (Fla) positively contributed to PC2, while DPPH, phenolic (Phe) and carotenoids (Caro) negatively correlated to PC2. Strong positive correlations were found between dry weight (DW), soluble sugar (Sugar) and vitamin C (VC), as their angles were less than 90°.

#### 2.1.7. Glucosinolate Content

In this study, five aliphatic glucosinolates (GSs) and four indole GSs of cabbage and Chinese kale microgreens were determined by HPLC (Figure 5). Aliphatic GSs include progoitrin (PRO), glucoraphanin (RAA), sinigrin (SIN), gluconapin (NAP) and glucobrassicanpin (GBN). Indole GSs include 4-hydroxygiucobrassicin (4OH), glucobrassicin (GBC), 4-methoxyglucobrassicin (4ME) and neoglucobrassicin (NEO). The total aliphatic GS content of cabbage and Chinese kale microgreens was significantly higher than total indole GS content. The total aliphatic GSs and indole GSs in cabbage microgreens was 84.4% and 15.6%, respectively (Figure 6A), and in Chinese kale microgreens was 92.3% and 7.7%, respectively (Figure 6B).

In cabbage microgreens, the SIN content was the highest, accounting for 76.2% of the total GSs, while the content of GBC, PRO and RAA accounted for 3.1%, 2.4% and 0.3%, respectively. In Chinese kale microgreens, the NAP content was the highest, accounting for 51.0% of the total GSs, while the content of PRO, SIN, GBC and RAA accounted for 18.8%, 8.2%, 1.6% and 0.4%, respectively.

For cabbage microgreens, the contents of total indole GSs, PRO, GBC, and NEO were significantly lower at 70 μmol·m^−2^·s^−1^. At a light intensity of 90 μmol·m^−2^·s^−1^, NAP content significantly increased (Figure 6A,C). For Chinese kale microgreens, the contents of total indole GSs, 4OH and 4ME were lower at 30 μmol·m^−2^·s^−1^. At 70 μmol·m^−2^·s^−1^, the content of total GSs and aliphatic GSs were the lowest (Figure 6B,D).

The contents of aliphatic GSs, such as PRO, NAP and GBN, in Chinese kale microgreens were significantly higher than those in cabbage microgreens, but the content of SIN was lower than that in cabbage microgreens. The contents of indole GSs, such as 4OH and GBC, in Chinese kale microgreens were significantly higher than those in cabbage microgreens (Figure 6C,D). The total GS content in Chinese kale was twice that of cabbage.

### 2.2. Effects of Photoperiod on Growth and Quality in Cabbage and Chinese Kale Microgreens

#### 2.2.1. Growth and Biomass

The hypocotyl length increased in cabbage microgreens under prolonged light exposure up to 14 h·d^−1^, with no significant differences observed among longer treatments. There was no significant difference in hypocotyl length of Chinese kale microgreens among different treatments. The dry weight and fresh weight of cabbage microgreens reached a maximum under the 20-h·d^−1^ photoperiod, and those of Chinese kale reached the maximum under the 14-h·d^−1^ photoperiod (Table 4).

#### 2.2.2. Pigment Content

The contents of chlorophyll a, chlorophyll b, total chlorophyll and carotenoid in cabbage and Chinese kale microgreens were the highest at 16 h·d^−1^ (Table 5) and lowest at 12 h·d^−1^. There was higher chlorophyll content in cabbage microgreens and higher carotenoid content in Chinese kale microgreens.

#### 2.2.3. Content of Soluble Protein, Soluble Sugar, Vitamin C and Nitrate

There was no significant difference in soluble sugar content among treatments in cabbage microgreens (Figure 7A). The soluble protein content increased with the photoperiod until 16 h·d^−1^, with no significant difference between 16, 18 and 20 h·d^−1^ (Figure 7B). The vitamin C content was the lowest at 16 h·d^−1^, with no significant difference among the other treatments (Figure 7C). Nitrate content varied significantly among treatments and was the lowest at 12 h·d^−1^ (Figure 7D).

For Chinese kale microgreens, neither soluble sugar nor soluble protein contents were significantly different among treatments (Figure 6B and Figure 7A). The contents of vitamin C and nitrate were the lowest at 14 h·d^−1^, with no significant differences among the other treatments (Figure 6D and Figure 7C ).

There were higher contents of soluble protein, vitamin C and nitrate in cabbage microgreens, and higher soluble sugar content in Chinese kale microgreens.

#### 2.2.4. Antioxidant Content and Capacity

There was no significant difference in DPPH and FRAP among cabbage microgreens under different photoperiod treatments (Figure 8). The contents of polyphenol and flavonoid were the lowest for the 20-h·d^−1^ photoperiod. For Chinese kale microgreens, DPPH and contents of polyphenol and flavonoid were not significantly different among photoperiod treatments, while FRAP was the lowest for the 16-h·d^−1^ photoperiod.

There was a higher flavonoid content in Chinese kale microgreens compared to cabbage microgreens.

#### 2.2.5. Heatmap Analysis

The heatmap presents the effects of different photoperiods on cabbage and Chinese kale microgreens. The hypocotyl length, fresh weight, dry weight, pigment and soluble protein contents of cabbage microgreens were the smallest under the 12-h·d^−1^ photoperiod, while the hypocotyl length, pigment, polyphenol, nitrate and flavonoid contents were the highest under the 16-h·d^−1^ photoperiod. (Figure 9A).

The heat map (Figure 9B) shows that the contents of flavonoids, soluble proteins, soluble sugars, polyphenols and vitamin C, as well as the FRAP value of Chinese kale microgreens were highest under the 12-h·d^−1^ photoperiod, while the pigment content as well as the dry and fresh weight were highest under the 16-h·d^−1^ photoperiod.

#### 2.2.6. Multivariate Principal Component Analysis

The global comparison of the correlations among phytochemicals in cabbage and Chinese kale microgreens grown under different photoperiods was generated by principal component analysis (PCA) (Figure 10).

For the PCA for cabbage microgreens (Figure 10A), the first four principal components (PCs) were associated with eigenvalues higher than 1 and explained 88.7% of the cumulative variance, with PC1 accounting for 39.1%, PC2 for 20.7%, PC3 for 12.4%, PC4 for 8.9% and PC5 for 7.7% (Table 6). PC1 was positively correlated to dry weight (DW), chlorophylls (Chl a, Chl b, and Total chl), FRAP, carotenoids (Caro), phenolic (Phe), soluble protein (Pro), flavonoids (Fla), vitamin C (VC), hypocotyl length (HL), nitrate (Nit) and fresh weight (FW). Additionally, DPPH, flavonoids (Fla), hypocotyl length (HL), nitrate (Nit), FRAP, chlorophyll a (Chl a), soluble protein (Pro) and phenolic (Phe) positively contributed to PC2. Strong positive correlations were found between dry weight (DW), soluble protein (Pro), FRAP, phenolic (Phe) and vitamin C (VC), as their angles were less than 90°.

For the PCA for Chinese kale microgreens (Figure 10B), the first three principal components (PCs) were associated with eigenvalues higher than 1 and explained 79.2% of the cumulative variance, with PC1 accounting for 35.9%, PC2 for 15.1%, PC3 for 11.2%, PC4 for 9.9% and PC5 for 7.1% (Table 6). PC1 was positively correlated to fresh weight (FW), dry weight (DW), chlorophylls (Chl a, Chl b, and Total chl), soluble sugar (Sugar), nitrate (Nit), and DPPH. Additionally, phenolic (Phe), hypocotyl length (HL), flavonoids (Fla), FRAP, DPPH, fresh weight (FW), dry weight (DW), soluble protein (Pro) and nitrate (Nit) positively contributed to PC2. Strong negative correlations were found between dry weight (DW), soluble sugar (Sugar) and vitamin C (VC), as their angles were less than 90°.

#### 2.2.7. Glucosinolate Content

In this study, the glucosinolates (GSs) in cabbage and Chinese kale microgreens included five aliphatic GSs and four indole GSs (Figure 5).

The aliphatic GS content in cabbage microgreens accounted for 79% of the total GS, and the indole GS content accounted for 21% (Figure 11A). Among them, SIN had the highest content, accounting for 77.9% of total GS, and GBC, PRO and RAA accounted for 6.7%, 0.5% and 0.1%, respectively. Under the 12-h·d^−1^ photoperiod, the GS contents were the lowest, and significantly lower in PRO, RAA, SIN, 4ME and NEO compared to the other treatments (Figure 11C).

The aliphatic GS content in Chinese kale microgreen accounted for 92% of the total GS, and the indole GS content accounted for 8% (Figure 11B). Among them, the NAP content was the highest, accounting for 52.1% of total GS, and PRO, SIN, GBC and RAA accounted for 17.1%, 8.5%, 1.2% and 1.2%, respectively. Under the 16-h·d^−1^ photoperiod, the GS contents were the lowest, and the contents of SIN, NAP, 4OH, GBN and NEO were significantly decreased compared with the other treatments (Figure 11D).

Aliphatic GSs were generally significantly higher in Chinese kale microgreens, particularly PRO, RAA, NAP and GBN, but SIN was significantly higher in cabbage microgreen. There were higher NEO contents in cabbage microgreens, and higher 4OH contents in Chinese kale microgreens.

## 3. Discussion

### 3.1. Effects of Light Intensity and Photoperiod on Growth of Cabbage and Chinese Kale Microgreens

Hypocotyls are one of the main edible parts of sprouts and microgreens, and are significantly affected by artificial lights [12]. Compared with dark conditions, hypocotyl length (HL) of soybean sprouts under 100 μmol·m^−2^·s^−1^ light intensity was significantly reduced by 16% [13]. Light intensity also plays a vital role in hypocotyl growth. HL in sprouts of alfalfa, broccoli, clover, kohlrabi, radish and red radish significantly decreased with blue light intensity increasing from 12.5 to 100 µmol·m^−2^·s^−1^ [9]. Similarly, the HL of kohlrabi, mizuna and mustard decreased progressively as light intensity increased [14]. HL of kohlrabi grown under R_84_:FR_7_:B_9_ decreased 32% as light intensity increased from 105 to 315 µmol·m^−2^·s^−1^ [14]. In this study, increasing light intensity from 30 to 90 μmol·m^−2^·s^−1^ caused HL in cabbage and Chinese kale microgreens to decrease by 19% and 24%, respectively (Table 1). It is well known that gibberellin (GA) plays a pivotal role in the regulation of shoot elongation [15]. Increased light intensity could reduce endogenous GA content in *Brassica* seedlings, causing inhibition of hypocotyl elongation [16]. The lower HL of microgreens under increased light intensity might be due to lower GA levels in the microgreens.

On the other hand, the fresh weight of *Brassica* microgreens increased 34% as light intensity increased from 105 to 315 μmol·m^−2^·s^−1^ [14]. The length and fresh weight of tartary buckwheat sprouts can be affected by light, even if the intensity is very low (50 μmol·m^−2^·s^−1^) [17]. The 50 μmol·m^−2^·s^−1^ treatment increased fresh weight and hypocotyl length of broccoli microgreens [7]. In this study, the dry and fresh weights of cabbage and Chinese kale microgreens reached the maximum at 70 μmol·m^−2^·s^−1^ (Table 1). It is worth mentioning that the photosynthetic pigment contents of the two microgreens also reach the maximum at 70 μmol·m^−2^·s^−1^. A 250 µmol·m^−2^·s^−1^, 20% blue and 80% red LED light has been shown to significantly increase total chlorophyll content of broccoli microgreens [18], and 90 μmol·m^−2^·s^−1^ white light enhanced chlorophyll content of broccoli microgreens [7]. A strong positive correlation between dry weight and photosynthetic pigment content was confirmed in cabbage and Chinese kale microgreens (Figure 4A,B). At 70 μmol·m^−2^·s^−1^, photosynthesis produced more carbohydrates, resulting in higher DW. Increased carotenoid contents were found under higher light intensity (>50 µmol·m^−2^·s^−1^) (Table 2). Carotenoids could play an essential photoprotective role by reducing photooxidative damage to the photosynthetic apparatus due to excess light intensity [19].

The role of photoperiods in the regulation of microgreens growth cannot be ignored. The dry weight and fresh weight of hydroponic kale linearly increased with the extension of the light cycle (from 6 to 24 h) [20]. As the photoperiod increased from 8 h to 20 h, the total fresh weight increased significantly in two amaranth microgreens (red amaranth and leafy vegetable amaranth) [21]. In this study, the dry and fresh weight of cabbage microgreens reached the maximum at the 20-h photoperiod (though the size at 20 h was not significantly different from that at 14 h). The fresh weight of Chinese kale microgreens reached the maximum at the 14-h photoperiod (Table 4). Higher pigment content was found at the 14-h·d^−1^ photoperiod in cabbage and Chinese kale microgreens, which might attribute to higher dry and fresh weight.

Microgreens treated with photoperiods of 8, 12, 16 and 20 h·d^−1^ showed that 20 h·d^−1^ reduced chlorophyll and carotenoid contents in amaranth seedling [21]. In this study, the contents of chlorophyll and carotenoid in cabbage and Chinese kale microgreens increased with photoperiod until reaching a maximum at 16 h·d^−1^, and then gradually decreased (Table 5). The pigment content could reach the maximum in the optimal photoperiod, while excessive photoperiods might mediate inhibition of photosynthetic activities, leading to biomass reduction.

### 3.2. Effects of Light Intensity and Photoperiod on Nutrition of Cabbage and Chinese Kale Microgreens

The lowest protein content in broccoli microgreens was observed at the higher light intensities (90 μmol·m^−2^·s^−1^) [7]. In cabbage microgreens, the soluble protein contents were highest at 90 μmol·m^−2^·s^−1^, whereas the Chinese kale microgreens reached their maximum at 70 μmol·m^−2^·s^−1^ and decreased significantly at 90 μmol·m^−2^·s^−1^ (Figure 1B). The influence of light on protein content might be species- and cultivars-dependent, and the mechanism needs further research.

Vitamin C is a cofactor for many enzymes and an essential health-promoting compound. Artificial light plays a positive role in promoting the vitamin C content of sprouts and microgreens. The vitamin C content in Chinese kale sprouts at 30 μmol·m^−2^·s^−1^ was higher than in the dark [22]. In this study, higher vitamin C contents were found in cabbage microgreens under 90 μmol·m^−2^·s^−1^ and Chinese kale microgreens under 70 μmol·m^−2^·s^−1^; higher photosynthetic pigment contents were also found under these two light intensities (Figure 1C). Vitamin C and carotenoids are important antioxidants, which in some plant species could reduce or clear reactive oxygen species (ROS) induced by high light intensity [23]. A strong positive correlation of vitamin C and photosynthetic pigments was confirmed in cabbage and Chinese kale microgreens (Figure 4).

The nitrate concentrations of kohlrabi, mustard, red pak choi and tatsoi were the highest under the lowest light (110 μmol·m^−2^·s^−1^) and decreased with increasing light [24]. From 100 to 800 µmol·m^−2^·s^−1^, increased light intensity gradually reduced the nitrate contents in lettuce [25]. In this study, lower nitrate contents were observed in higher light intensity (90 μmol·m^−2^·s^−1^) in cabbage and Chinese kale microgreens. This could be attributed to higher nitrate reductase activity induced by higher light intensity, resulting in the decrease of nitrate accumulation [26].

Lettuce under longer photoperiods developed higher contents of vitamin C, soluble sugar and soluble protein, with all of these reaching their maximum at 16 h·d^−1^ [27]. Similarly, in this study the soluble protein content in cabbage microgreens increased with the prolonged photoperiod and reached a maximum at 16 h·d^−1^ (Figure 7B). The prolongation of the photoperiod increased photosynthetic time and thus increased carbon assimilation. Differing from cabbage, Chinese kale microgreens displayed no significant differences in vitamin C, soluble sugar or soluble protein contents for the different photoperiods (Figure 7A–C). This indicates that the effects of photoperiod on plant metabolism largely depends on cultivars. Additionally, a shorter photoperiod (<18 h·d^−1^) reduced nitrate content in cabbage and Chinese kale microgreens (Figure 7D).

### 3.3. Effects of Light Intensity and Photoperiod on Antioxidant Content and Activity in Cabbage and Chinese Kale Microgreens

Plants respond to environmental stress by producing antioxidants as a defense mechanism. Light environment affects the total phenolic content of canola sprouts, Chinese kale seedlings and pea sprouts [28,29,30]. In this study, increasing light intensity resulted in significant increases in polyphenol content in cabbage and Chinese kale microgreens. Cabbage and Chinese kale microgreens exhibited higher antioxidant activity in response to increased light intensity, which was mainly due to their higher polyphenol and flavonoid levels (Figure 2). Increased light enhanced the accumulation of flavonoids in *Matteuccia struthiopteris* [31]. Compared to dark conditions, 30 μmol·m^−2^·s^−1^ blue and red LED light increased the total flavonoid content in pea sprouts [30]. Our study showed that increasing light intensity resulted in significant increases in flavonoid content in cabbage and Chinese kale microgreens, possibly due to higher light intensity inducing the expression of phenylalanine ammonia lyase, which is involved in the synthesis of phenolic and flavonoid [22,32].

In this study, cabbage microgreens showed the lowest FRAP value under the 20-h·d^−1^ photoperiod, mainly due to the low contents of polyphenol and flavonoid (Figure 8B–D). The total antioxidant capacity of amaranth reached the maximum under a 16-h·d^−1^ photoperiod and then decreased under a 20-h·d^−1^ photoperiod [21]. There was a reduction in total polyphenol contents in five leafy vegetables (red amaranth, green amaranth, red beet, swiss chard and red spinach) after the illumination time extended beyond a threshold photoperiod (>12 h·d^−1^) [33]. Thus, the length of photoperiod that provides maximal antioxidant capacity might vary among crop species.

### 3.4. Effects of Light Intensity and Photoperiod on Glucosinolate Content in Cabbage and Chinese Kale Microgreens

Glucosinolates (GSs) and their breakdown products, isothiocyanates, play important roles in crop flavor and plant defense, and as anticancer agents in the human body [34]. The anabolism of GSs is affected by the light environment [35]. In this study, nine GSs were identified in cabbage and Chinese kale microgreens (Figure 5), and the GS contents in Chinese kale were significantly higher than those in cabbage under the different light intensities and photoperiods (Figure 6 and Figure 11). In broccoli microgreens, the contents of PRO, SIN and GBN decreased significantly when the light intensity increased from 30 to 50 μmol·m^−2^·s^−1^ [18]. However, although the contents of PRO, SIN, GBC and NEO in cabbage microgreens decreased significantly at 70 μmol·m^−2^·s^−1^, the changed intensity did not strikingly affect the total GS content in either the cabbage or the Chinese kale microgreens (Figure 6). GS accumulation was affected by different light intensities and photoperiods. It has been reported that increases in light time could promote GS accumulation in watercress [36]. However, different photoperiods (0, 8, 12 and 16 h) did not affect the GS content in Chinese kale sprouts [11]. SIN is the most abundant GS (81.08%) in cabbage microgreens (Figure 11) and obviously increased with the photoperiod, which attributed to the higher total GS content for the higher photoperiod. These results indicate the complex relationship between the light environment and glucosinolates biosynthesis.

## 4. Materials and Methods

### 4.1. Plant Materials and Treatments

The experiments were conducted in the artificial-lighting plant factory of South China Agricultural University. Seeds of cabbage (‘Huafeng Zhonggan 11′) and Chinese kale (‘Sijicutai’) were soaked in water for 4 h, then sown onto plastic trays (38 cm × 28 cm × 2 cm), kept in the dark for 3 days, then transferred to a cultivation shelf with red/blue/green LED light (Chenghui Equipment Co., Ltd., Guangzhou, China). Spectral data were measured using a lighting passport (ALP-01, Asensetek, Taiwan) (Figure 12A).

Experiment 1: There were four light intensity treatments of 30, 50, 70 or 90 μmol·m^−2^·s^−1^, with 12 h irradiation from 8:00 to 20:00 every day.

Experiment 2: An LED light intensity at 70 μmol·m^−2^·s^−1^ was provided for five photoperiod treatments of 12, 14, 16, 18 or 20 h·d^−1^.

The microgreen shoots were collected and sampled on the eighth day after they were sown. The samples were ground with mortar and pestle to fine powder in liquid nitrogen and stored in a freezer at −40 °C. For each treatment, three replicates were taken for analysis. The experimental bench is shown in Figure 12B.

### 4.2. Growth Determination

Twenty microgreens were randomly sampled from each treatment. Hypocotyl lengths were measured by a ruler. Fresh weight (FW) was determined by an electronic analytical balance. Fresh microgreens were deactivated in an oven at 105 °C for 2 h, then oven dried at 75 °C to constant weight to obtain the dry weight (DW).

### 4.3. Phytochemical Determination

#### 4.3.1. Chlorophyll (Chl) and Carotenoid Contents

Fresh samples of cotyledon of microgreens (0.2 g) were soaked in 6.0 mL 95% ethanol and incubated at 25 °C in the dark for 24 h. The extract solution absorbance was determined with a UV-spectrophotometer (Shimadzu UV^−1^ 6A, Shimadzu, Corporation, Kyoto, Japan) at 665 nm (A_665_), 649 nm (A_649_) and 470 nm (A_470_). The pigment contents were calculated as follows [37]: Chl a concentration (mg/g FW) = (13.36 × A_665_ − 5.19 × A_665_) × 6 mL/1000/0.2 g; Chl b concentration (mg/g FW) = (27.43 × A_649_ − 8.12 × A_665_) × 6 mL/1000/0.2 g; Carotenoid concentration (mg/g FW) = (1000 × A_470_ − 2.13 × C_a_ − 97.64 × C_b_)/209 × 6 mL/1000/0.2 g.

#### 4.3.2. Soluble Protein Content

Determination of soluble protein content was performed by Coomassie brilliant blue G-250 staining [38]. Fresh plant tissue (0.5 g) was mixed with 8 mL distilled water and then centrifuged at 3000× *g* for 10 min at 4 °C. The supernatant (0.2 mL) was combined with 0.8 mL distilled water and 5.0 mL Coomassie brilliant blue G-250 solution (0.1 g·L^−1^). After 5 min, the absorbance was measured at 595 nm using a UV- spectrophotometer.

#### 4.3.3. Soluble Sugar Content

Soluble sugar content was determined by anthrone-sulfuric acid colorimetry [39]. Fresh frozen tissue (0.5 g) was mixed with 5.0 mL 80% ethanol in a test tube and kept in an 80 °C water bath for 40 min. Another 5.0 mL 80% ethanol was added, and the test tube was returned to the 80 °C water bath for another 40 min, then the solution was filtered by a funnel with double filter papers. The filtered solution was collected in a 10-mL volumetric flask, cooled to 25 °C, and then 80% ethanol was added until the volume was 10 mL. Later, the filtered solution (0.2 mL) and deionized water (0.8 mL) were mixed in a 10 mL test tube; 0.5 mL anthrone ethyl acetate reagent and 5.0 mL concentrated sulfuric acid were added, mixed with vortex, and then placed in a boiling water bath for 10 min. After cooling to ambient temperature, the solution was measured at 625 nm by a UV-spectrophotometer.

#### 4.3.4. Flavonoid Content

The flavonoid content was determined by an aluminum nitrate method [40]. Fresh frozen tissue (0.5 g) was extracted by 8.0 mL absolute ethanol. Next 1.0 mL extract was mixed with 0.7 mL 5% sodium nitrite solution in a 10 mL test tube for 5 min. Then 0.7 mL 5% aluminum nitrate was added to the mixture for 6 min. A total of 5.0 mL 5% sodium hydroxide solution was added and reacted at 25 °C. The absorption at 510 nm was measured by a UV-spectrophotometer.

#### 4.3.5. Vitamin C Content

Vitamin C content was determined by molybdenum blue spectrophotometry [41]. Fresh frozen tissue (0.5 g) was homogenized with 25 mL 0.05 moL/L oxalic acid solution (*w*/*v*) in a volumetric flask. Then the solution was filtered by a funnel with double filter papers. Next 10.0 mL supernatant was mixed with 1.0 mL metaphosphoric acid solution (*w*/*v*), 2.0 mL 5% sulfuric acid solution (*v*/*v*) and 4.0 mL 5% ammonium molybdate solution (*w*/*v*). The supernatants were mixed well and sat still for 15 min, then measured at 705 nm by a UV-spectrophotometer, using oxalic acid as a blank.

#### 4.3.6. Polyphenol Content

The polyphenol content was determined using the Folin–Ciocalteu assay [42]. Fresh frozen tissue (0.5 g) was extracted with 8 mL alcohol. The homogenate was allowed to stand for 30 min and then centrifuged at 3000× *g* at 4 °C for 10 min. The supernatant (1 mL) was then mixed with 0.5 mL Folin phenol and 11.5 mL 26.7% sodium carbonate; 7.0 mL distilled water was then added to the mixture. The absorbance was measured at 510 nm using a UV-spectrophotometer.

#### 4.3.7. DPPH Radical-Scavenging Rate

The 2, 2-diphenyl^−1^-picrylhydrazyl (DPPH) radical-scavenging rate was determined following Tadolini et al. [43]. The sample extract (2.0 mL) was mixed with 2.0 mL DPPH solution (0.0080 g DPPH in 100 mL alcohol) and the absorbance of the mixture was determined at 517 nm using a UV-spectrophotometer.

#### 4.3.8. Ferric-Reducing Antioxidant Power

The ferric-reducing antioxidant power (FRAP) was determined following Benzie and Strain [44]. The sample solution (0.4 mL), extracted following the same method as for polyphenols, was mixed with 3.6 mL solution containing 0.3 mol·L^−1^ acetate buffer, 10 mmol·L^−1^ 2, 4, 6-tripyridyl-S-triazine (TPTZ) and 20 mmol·L^−1^ FeCl_3_ at a 10:1:1 ratio (*v*/*v*/*v*) for 10 min at 37 °C. The FRAP was determined at 593 nm by a spectrophotometer.

#### 4.3.9. Nitrate Content

The nitrate content was determined by UV spectrophotometry [45]. Fresh plant tissue (1.0 g) was homogenized in 10 mL distilled water and heated in a boiling water bath for 30 min. The homogenate was filtered into a volumetric flask. Then 0.1 mL sample solution was mixed with 0.4 mL 5% salicylic and sulfuric acid and 9.5 mL 8% NaOH. The nitrate content was determined using a UV-spectrophotometer at 410 nm.

#### 4.3.10. Glucosinolate Content Determination

Determination of glucosinolates was performed with HPLC referring to Gao et al. [7]. The frozen-dried sample was extracted with methanol, and then the extracts were purified and desulfurized with the ion-exchange method. The glucosinolates were separated and identified by HPLC (Waters Alliance e2695, Milford, MA, USA). A 5 µm C18 column (Waters, 250 mm length, 4.6 mm diameter) was used for glucosinolate separation. Elution was performed with mobile phase A (water, 18.2 MΩ·cm resistance) and mobile phase B (acetonitrile). The optimum column temperature was 30 °C. At a flow rate of 1.0 mL/min, the gradient conditions were set as follows: solvent A volume at 100% for 0 to 32 min, 80% for 32 to 38 min, and solvent B volume at 100% for 38 to 40 min. The time for elution was at 42–50 min. The detector monitored glucosinolates at 229 nm. The individual glucosinolates were identified according to their HPLC retention times and our database, and quantified with sinigrin as an internal reference with their HPLC areas and relative response factors.

### 4.4. Statistical Analysis

Data were expressed as mean ± standard error (*n* = 3 replicates) and analyzed by two-way analysis of variance (ANOVA) using SPSS 22.0 software (Chicago, IL, USA). Means were compared using Duncan’s test, and differences were considered significant at *p* < 0.05. Figures were drawn using Origin 2018 software (Origin Lab, Northampton, MA, USA). Heat maps were generated using TBtools [46].

## 5. Conclusions

The suitable light intensity and photoperiod could promote growth and improve the nutrition of cabbage and Chinese kale microgreens. For cabbage and Chinese kale microgreens, 90 μmol·m^−2^·s^−1^ and 70 μmol·m^−2^·s^−1^ light intensities, respectively, were beneficial to improve biomass and nutrients, such as chlorophyll, carotenoids, soluble sugars, soluble proteins and vitamin C, and antioxidant capacity. The 14–16-h·d^−1^ photoperiod could improve biomass, nutritional quality and some antioxidant capacity. Future studies should focus on the combined treatments of proper light intensity and photoperiod for higher phytochemical contents without inhibiting the growth of microgreens.

## Figures and Tables

**Figure 1 molecules-27-00883-f001:**
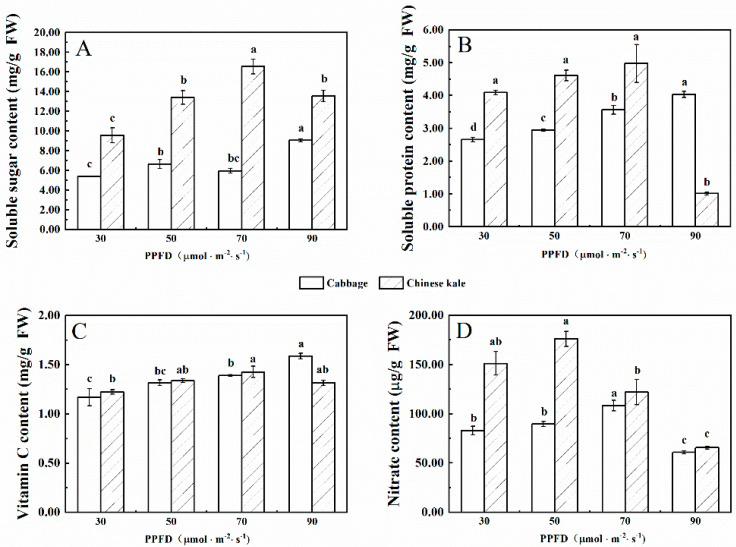
The contents of soluble sugar (**A**), soluble protein (**B**), vitamin C (**C**) and nitrate (**D**) in cabbage and Chinese kale microgreens under different light intensities treatments. Different letters (a–d) on the bar plots indicate significant difference at *p* < 0.05 using one-way analysis of variance with Duncan’s multiple-range test.

**Figure 2 molecules-27-00883-f002:**
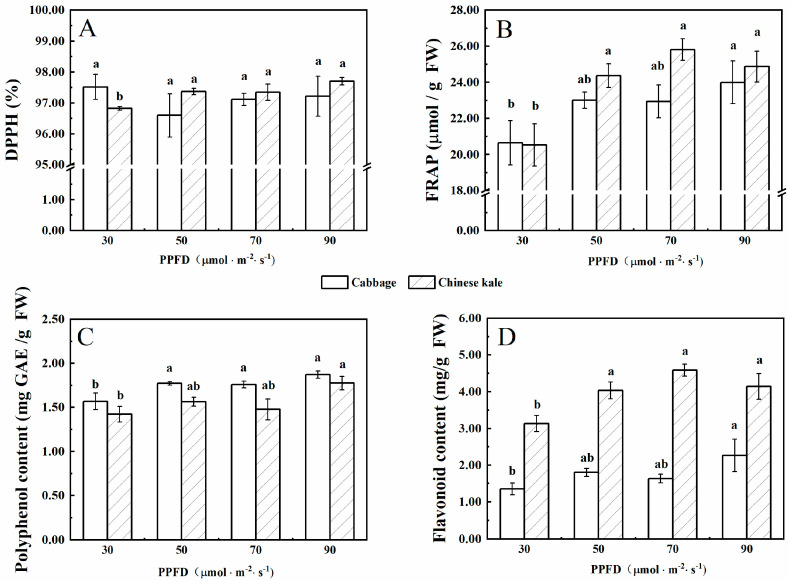
The contents of DPPH (**A**), FRAP (**B**), polyphenol (**C**), and flavonoid (**D**) in cabbage and Chinese kale microgreens under different light intensities treatments. Different letters (a,b) on the bar plots indicate significant difference at *p* < 0.05 using one-way analysis of variance with Duncan’s multiple-range test.

**Figure 3 molecules-27-00883-f003:**
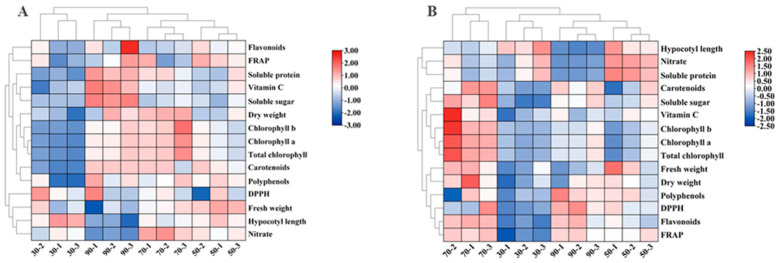
The heat map shows the differences in growth and quality of cabbage (**A**) and Chinese kale (**B**) microgreens under various light intensities. The shades of red and blue indicate the relative value of the parameter.

**Figure 4 molecules-27-00883-f004:**
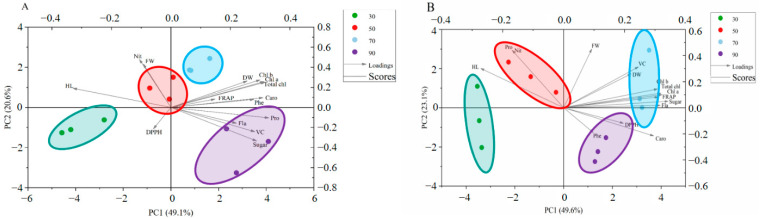
Multivariate principal component analysis showing how different light intensities enhanced quality of cabbage (**A**) and Chinese kale (**B**): DW (dry weight); Chl a (Chlorophyll a); Chl b (Chlorophyll b); Total chl (total chlorophyll); Caro (carotenoids); Phe (phenolic); Pro (soluble protein); Fla (flavonoids); VC (vitamin C); Sugar (soluble sugar); HL (hypocotyl length); Nit (nitrate); FW (fresh weight).

**Figure 5 molecules-27-00883-f005:**
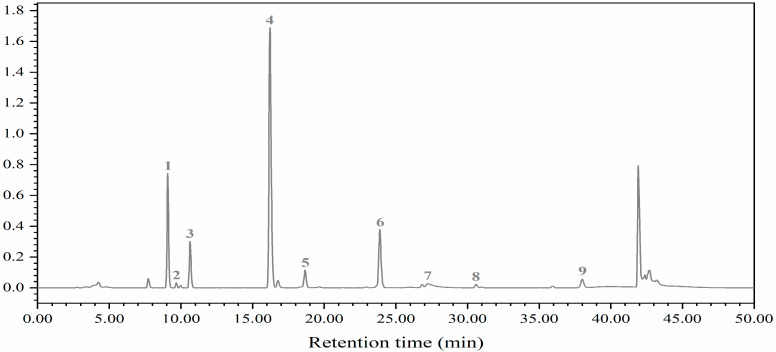
HPLC chromatogram of glucosinolates: 1, progoitrin (PRO); 2, glucoraphanin (RAA); 3, sinigrin (SIN); 4, gluconapin (NAP); 5, 4-hydroxygiucobrassicin (4OH); 6, glucobrassicanpin (GBN); 7, glucobrassicin (GBC); 8, 4-methoxyglucobrassicin (4-ME); 9, neoglucobrassicin (NEO).

**Figure 6 molecules-27-00883-f006:**
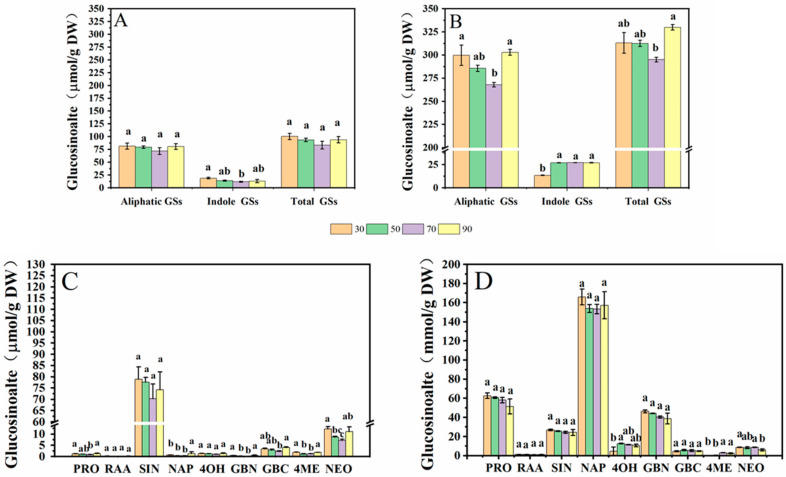
Effects of different light intensities on the GS content of cabbage (**A**,**C**) and Chinese kale (**B**,**D**) microgreens. Different letters (a–c) on the bar plots indicate significant difference at *p* < 0.05 using one-way analysis of variance with Duncan’s multiple-range test.

**Figure 7 molecules-27-00883-f007:**
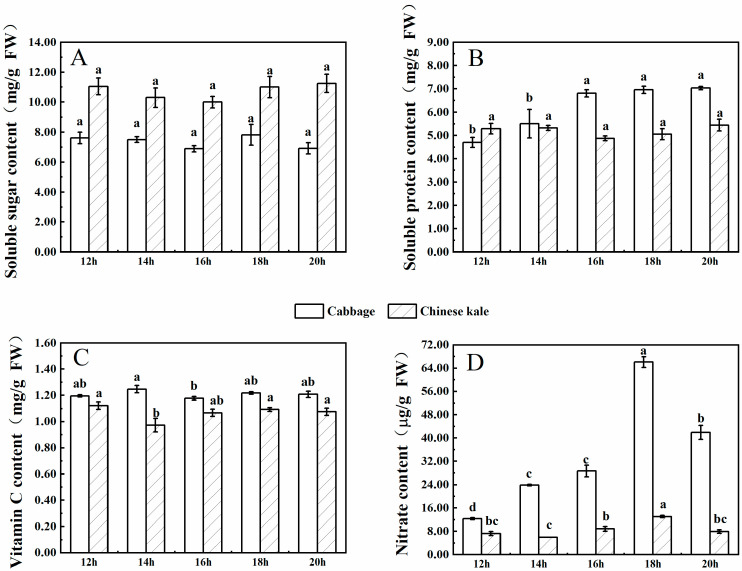
The contents of soluble sugar (**A**), soluble protein (**B**), vitamin C (**C**) and nitrate (**D**) in cabbage and Chinese kale microgreens under different photoperiods treatments. Different letters (a–d) on the bar plots indicate significant difference at *p* < 0.05 using one-way analysis of variance with Duncan’s multiple-range test.

**Figure 8 molecules-27-00883-f008:**
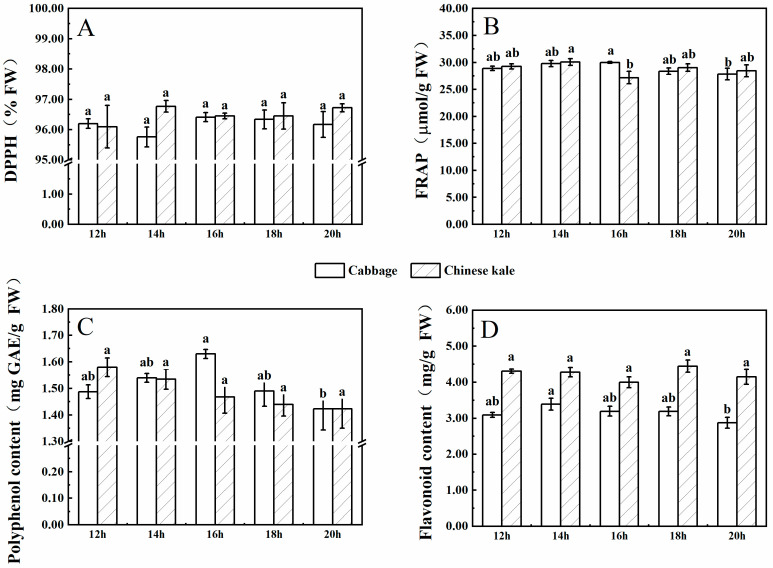
The contents of DPPH (**A**), FRAP (**B**), polyphenol (**C**), and flavonoid (**D**) in cabbage and Chinese kale microgreens under different photoperiod treatments. Different letters (a,b) on the bar plots indicate significant difference at *p* < 0.05 using one-way analysis of variance with Duncan’s multiple-range test.

**Figure 9 molecules-27-00883-f009:**
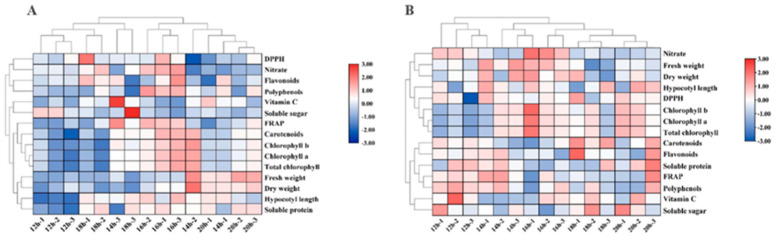
The heatmap shows the difference in growth and quality of cabbage (**A**) and Chinese kale (**B**) microgreens under different photoperiods. The shades of red and blue indicate the relative value of the parameter.

**Figure 10 molecules-27-00883-f010:**
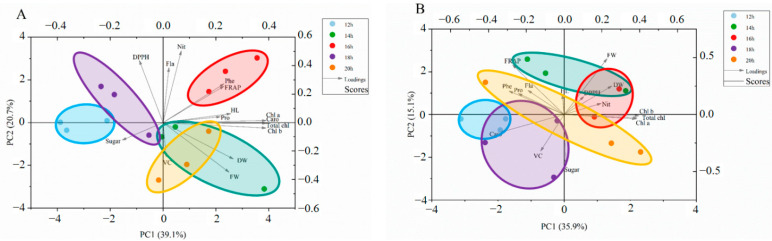
Multivariate principal component analysis showing how different photoperiods enhanced quality of cabbage (**A**) and Chinese kale (**B**): DW (dry weight), ChlA (Chlorophyll a), Chl B (Chlorophyll b), Total chl (total chlorophyll), Caro (carotenoids), Phe (phenolic), Pro (soluble protein), Fla (flavonoids), VC (vitamin C), Sugar (soluble sugar), HL (hypocotyl length), Nit (nitrate), FW (fresh weight).

**Figure 11 molecules-27-00883-f011:**
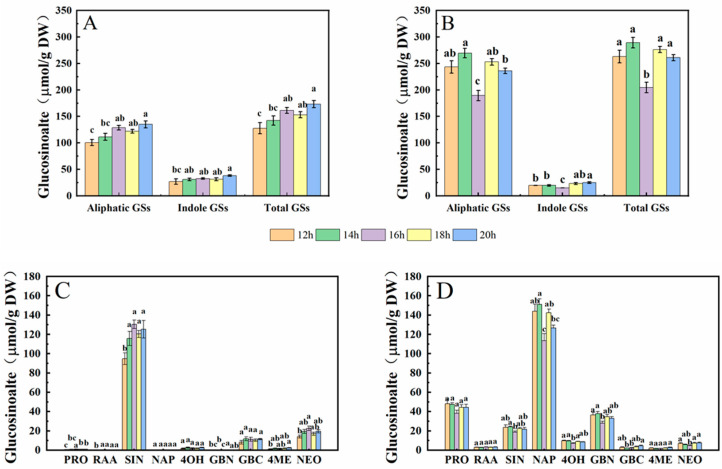
Effects of different photoperiods on the GS content of cabbage (**A**,**C**) and Chinese kale (**B**,**D**) microgreens. Different letters (a–c) on the bar plots indicate significant difference at *p* < 0.05 using one-way analysis of variance with Duncan’s multiple-range test.

**Figure 12 molecules-27-00883-f012:**
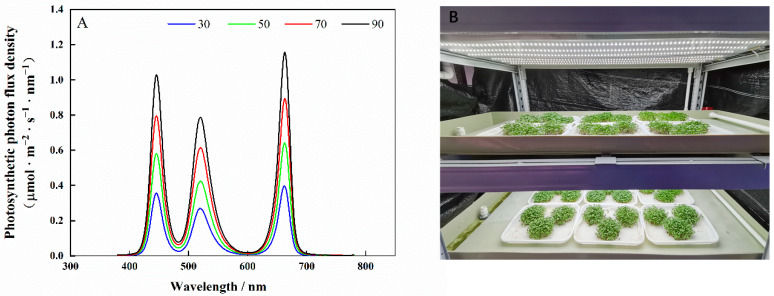
Spectra of light treatments (**A**) and photo of the experimental bench (**B**).

**Table 1 molecules-27-00883-t001:** Effects of light intensity on the growth of cabbage and Chinese kale microgreens.

Cultivars	PPFD μmoL·m^−2^·s^−1^	Hypocotyl Length cm	Fresh Weight (per Plant) mg	Dry Weight (per Plant) mg
Cabbage	30	4.83 ± 0.10 a	52.13 ± 1.43 b	2.66 ± 0.09 b
	50	4.70 ± 0.14 ab	57.75 ± 0.65 a	2.88 ± 0.12 b
	70	4.53 ± 0.04 b	54.25 ± 0.88 ab	3.21 ± 0.06 a
	90	4.07 ± 0.06 c	47.50 ± 1.65 c	2.95 ± 0.10 ab
Chinese kale	30	4.43 ± 0.08 a	36.88 ± 0.85 b	2.42 ± 0.06 b
	50	4.35 ± 0.20 a	42.63 ± 1.18 a	2.79 ± 0.08 a
	70	3.97 ± 0.06 b	43.63 ± 0.75 a	2.86 ± 0.11 a
	90	3.57 ± 0.10 c	36.75 ± 1.15 b	2.71 ± 0.07 a

Note: different lowercase letters indicate significant differences (*p* < 0.05) within the same column of the table.

**Table 2 molecules-27-00883-t002:** Effects of light intensity on the photosynthetic pigment content of cabbage and Chinese kale microgreens.

Cultivars	PPFD μmol·m^−2^·s^−1^	Chl a (mg/g)	Chl b (mg/g)	Total Chl (mg/g)	Carotenoid (mg/g)
Cabbage	30	0.25 ± 0.01 c	0.07 ± 0.02 c	0.32 ± 0.02 c	0.063 ± 0.004 b
	50	0.48 ± 0.04 b	0.15 ± 0.01 b	0.64 ± 0.05 b	0.110 ± 0.005 a
	70	0.67 ± 0.05 a	0.22 ± 0.02 a	0.90 ± 0.08 a	0.107 ± 0.009 a
	90	0.57 ± 0.03 ab	0.18 ± 0.01 ab	0.76 ± 0.04 ab	0.117 ± 0.000 a
Chinese kale	30	0.32 ± 0.02 b	0.10 ± 0.00 b	0.42 ± 0.02 b	0.075 ± 0.003 b
	50	0.34 ± 0.06 b	0.10 ± 0.02 b	0.44 ± 0.08 b	0.077 ± 0.010 b
	70	0.67 ± 0.05 a	0.22 ± 0.02 a	0.89 ± 0.08 a	0.102 ± 0.008 a
	90	0.44 ± 0.03 b	0.13 ± 0.01 b	0.58 ± 0.05 b	0.099 ± 0.003 a

Note: different lowercase letters indicate significant differences (*p* < 0.05) within the same column of the table.

**Table 3 molecules-27-00883-t003:** Eigenvalues, factor scores and contribution of the five principal component axes of cabbage and Chinese kale responses to different light intensities.

Cultivars	Principal Components	PC1	PC2	PC3	PC4	PC5
Cabbage	Eigenvalue	7.4	3.1	1.7	1.1	0.6
	Variance (%)	49.1	20.6	11.7	7.1	3.8
	Cumulation (%)	49.1	69.7	81.4	88.4	92.2
Chinese kale	Eigenvalue	7.4	3.5	2.4	0.7	0.3
	Variance (%)	49.6	23.1	16.0	5.0	2.3
	Cumulation (%)	49.6	72.7	88.7	93.7	96.0

**Table 4 molecules-27-00883-t004:** Effects of photoperiod on the growth of cabbage and Chinese kale microgreens.

Cultivars	Photoperiod h·d^−1^	Hypocotyl Length cm	Fresh Weight (per Plant) mg	Dry Weight (per Plant) mg
Cabbage	12	3.26 ± 0.11 b	62.88 ± 1.08 b	4.11 ± 0.14 bc
	14	3.93 ± 0.13 a	73.75 ± 1.89 a	4.64 ± 0.21 a
	16	4.01 ± 0.15 a	67.50 ± 1.18 b	4.47 ± 0.12 ab
	18	3.95 ± 0.07 a	62.63 ± 1.02 b	3.93 ± 0.11 c
	20	3.80 ± 0.16 a	77.50 ± 2.72 a	4.76 ± 0.16 a
Chinese kale	12	2.89 ± 0.15 a	55.33 ± 0.99 bc	4.22 ± 0.12 ab
	14	3.09 ± 0.14 a	62.78 ± 1.51 a	4.49 ± 0.15 ab
	16	3.08 ± 0.06 a	59.00 ± 1.53 b	4.56 ± 0.12 a
	18	2.99 ± 0.17 a	51.11 ± 1.55 d	4.11 ± 0.15 b
	20	3.12 ± 0.12 a	54.78 ± 0.68 cd	4.27 ± 0.10 ab

Note: different lowercase letters indicate significant differences (*p* < 0.05) within the same column of the table.

**Table 5 molecules-27-00883-t005:** Effects of photoperiod on the photosynthetic pigment content of cabbage and Chinese kale microgreens.

Cultivars	Photoperiod h·d^−1^	Chl a (mg/g)	Chl b (mg/g)	Total Chl (mg/g)	Carotenoid (mg/g)
Cabbage	12	0.66 ± 0.02 c	0.26 ± 0.01 c	0.92 ± 0.03 c	0.151 ± 0.005 b
	14	0.74 ± 0.03 ab	0.30 ± 0.01 ab	1.04 ± 0.04 ab	0.170 ± 0.007 a
	16	0.77 ± 0.02 a	0.31 ± 0.01 a	1.08 ± 0.02 a	0.175 ± 0.003 a
	18	0.68 ± 0.03 bc	0.27 ± 0.01 bc	0.94 ± 0.04 bc	0.158 ± 0.007 ab
	20	0.72 ± 0.01 abc	0.29 ± 0.01 abc	1.01 ± 0.02 abc	0.165 ± 0.003 ab
Chinese kale	12	0.47 ± 0.03 b	0.18 ± 0.01 b	0.65 ± 0.04 b	0.175 ± 0.010 b
	14	0.58 ± 0.04 ab	0.22 ± 0.02 ab	0.81 ± 0.06 ab	0.222 ± 0.019 ab
	16	0.68 ± 0.05 a	0.26 ± 0.02 a	0.94 ± 0.08 a	0.260 ± 0.024 a
	18	0.58 ± 0.04 ab	0.21 ± 0.02 ab	0.79 ± 0.06 ab	0.210 ± 0.016 ab
	20	0.65 ± 0.03 b	0.24 ± 0.01 a	0.89 ± 0.04 a	0.242 ± 0.014 a

Note: different lowercase letters indicate significant differences (*p* < 0.05) within the same column of the table.

**Table 6 molecules-27-00883-t006:** Eigenvalues, factor scores and contribution of the five principal component axes of cabbage and Chinese kale responses to different photoperiods.

Cultivars	Principal Components	PC1	PC2	PC3	PC4	PC5
Cabbage	Eigenvalue	5.9	3.1	1.9	1.3	1.1
	Variance (%)	39.1	20.7	12.4	8.9	7.7
	Cumulation (%)	39.1	59.8	72.1	81.0	88.7
Chinese kale	Eigenvalue	5.4	2.3	1.7	1.5	1.1
	Variance (%)	35.9	15.1	11.2	9.9	7.1
	Cumulation (%)	35.9	51.0	62.2	72.1	79.2

## Data Availability

Not applicable.

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
