# Peer review of "Light Intensity and Photoperiod Affect Growth and Nutritional Quality of Brassica Microgreens"

_molecules, 2022, doi:10.3390/molecules27030883_

Round 1

Reviewer 1 Report

This paper discusses the influence of photosynthetic photon flux density and photoperiod on the growth rate and the nutritional quality of bassica microgreens. The topic is relevant to scientific investigation and has significant practical importance for optimizing cultivation technologies in vertical farms. My major concern is about the reproducibility of measurement data presented in the manuscript.

Another concern is related to Figure 12.  Vertical axis should refer to a spectral quantity. There is a contradiction between vertical axis title and units used.

For the readers information and quantitative data about the uniformity of photon flux density would be useful, because this may be a source of error in the  experiments. I suggest to present PPFDmin / PPFDmax measured in the cultivation cells.

Detailed comments have been uploaded in a word file.

Author Response

Thank so much for your valuable and helpful comments and suggestions to improve our paper.Detailed reviewers have been uploaded in a word file. 

Reviewer 2 Report

The article needs some revision to be suitable for publication

-The abstract should be informative and include the main findings.

-Language should be revised. 

-Figures should be represented in higher resolution

- The introduction should be enriched with recent references (2016-2021)

  • Some important references might be added and discussed
  • 1- Evaluation of growth and nutritional value of Brassica microgreens grown under red, blue and green LEDs combinations. Physiol Plantarum, 169: 625-638. https://doi.org/10.1111/ppl.13083

Author Response

(The authors gave the same response as above.)

Reviewer 3 Report

Attached is a PDF file with a summary of my comments for this manuscript. 

Author Response

(The authors gave the same response as above.)

Round 2

Reviewer 1 Report

The title of the Y-axis in Figure 12 A should be Spectral Photosynthetic Photon Flux Density  (µmol·m-2 · m-2 · s-1 · nm-1).

The integral of the spectrum (the area under the curve) is equal to the PPFD. The legend provides the PPFD value for each curve in µmol·m-2 · m-2 · s-1 .  

Author Response

Thank so much for your valuable and helpful comments and suggestions to improve our paper. We have revised it based on your suggestion. Please see them in Figure 12 A)

Reviewer 2 Report

The revised version could be accepted

Author Response

Thank so much for your valuable and helpful comments and suggestions to improve our paper.

Reviewer 3 Report

I appreciate the authors' efforts to fortify the manuscript and address concerns brought up in the first round of review. I feel that the introduction serves a much more useful purpose in contextualizing the research that was performed in this study. I appreciate the addition of the photograph of the experimental setup as that is helpful for readers to visualize the design of the study.

As the author’s pointed out, there is no legal definition and microgreens can exist as either just cotyledons, or with cotyledons with fully expanded first true leaves. Given that these are distinct physiological stages, the authors should articulate the stage in which they sampled (e.g. “10 days after germination”) within the manuscript to make the study more reproducible to others as well as growers.

Regarding the carotenoids, I understand that the initial solvent composition was a typo. I’m still not convinced this was an appropriate approach. I saw the authors cited a study that compared carotenoid and chlorophyll spectral shifts in different solvent compositions and used their formulas to estimate carotenoid and chlorophyll content, but that paper was not outlining an extraction method for leaves. Other papers by that author (Lichtenthaler) emphasize the use of more nonpolar solvents and grinding leaves to completely extract fat soluble pigments from leaves. I’m still not clear where the authors are basing their 24 hour ethanol soaking method on since there isn’t a citation for that and if they developed that method, there are no reports of extraction efficiency. I pointed out that the carotenoid and chlorophyll values seemed to be a bit low compared to other microgreen publications and this observation might be related to an incomplete extraction. I think this issue needs to at least be acknowledged in the results and discussion somewhere.

The discussion has been fortified and better connected to the manuscript. I still feel like there are spots of inadequate connection between their findings and what has been published, but this draft is an improvement over the previous submission. I appreciate the inclusion of a concluding paragraph which helps tie up the study.

Overall, a big improvement. Other than the items I mentioned above, I would strongly encourage the authors to carefully proof read the manuscript for grammatical errors.

Author Response

Thank so much for your valuable and helpful comments and suggestions to improve our paper. I have uploaded responses in a word file.
